# Node Importance Identification for Temporal Networks Based on Optimized Supra-Adjacency Matrix

**DOI:** 10.3390/e24101391

**Published:** 2022-09-29

**Authors:** Rui Liu, Sheng Zhang, Donghui Zhang, Xuefeng Zhang, Xiaoling Bao

**Affiliations:** 1School of Information Engineering, Nanchang Hangkong University, 696 Fenghe South Avenue, Nanchang 330063, China; 2School of Foreign Language, Nanchang Hangkong University, 696 Fenghe South Avenue, Nanchang 330063, China

**Keywords:** temporal network, intra-layer relationship, inter-layer relationship, eigenvector-based centrality, node importance sorted list

## Abstract

The research on node importance identification for temporal networks has attracted much attention. In this work, combined with the multi-layer coupled network analysis method, an optimized supra-adjacency matrix (OSAM) modeling method was proposed. In the process of constructing an optimized super adjacency matrix, the intra-layer relationship matrixes were improved by introducing the edge weight. The inter-layer relationship matrixes were formed by improved similarly and the inter-layer relationship is directional by using the characteristics of directed graphs. The model established by the OSAM method accurately expresses the structure of the temporal network and considers the influence of intra- and inter-layer relationships on the importance of nodes. In addition, an index was calculated by the average of the sum of the eigenvector centrality indices for a node in each layer and the node importance sorted list was obtained from this index to express the global importance of nodes in temporal networks. The experimental results on three real temporal network datasets Enron, Emaildept3, and Workspace showed that compared with the SAM and the SSAM methods, the OSAM method has a faster message propagation rate and larger message coverage and better SIR and NDCG@10 indicators.

## 1. Introduction

Understanding the topological structure of complex networks is of fundamental importance for studying the dynamic behaviors of various systems [1]. The heterogeneous nature of a complex network determines the roles of each node in the network, which are quite different. Many mechanisms of complex networks such as spreading dynamics, cascading reactions, and network synchronization are highly affected by a tiny fraction of so-called important nodes [2]. Identifying important nodes plays a significant role in analyzing the characteristics of the net system and understanding the network structure and function [3]. There are many methods to identify the importance of nodes: degree centrality (DC) [4], betweenness centrality (BC) [5], closeness centrality (CC) [6], k-shell [7], Entropy Variation [2], and so on [8,9]. These methods are widely used in static networks, but they cannot be directly applied in temporal networks. This is because edges between nodes in temporal networks will appear and disappear intermittently over time [10].

In recent years, researchers have begun to explore the importance of nodes in temporal networks and have proposed a series of related methods [11]. Kim et al. [12] presented a simple yet effective model, the time-ordered graph, which reduces a dynamic network to a static one with directed flows, and defined the temporal versions of degree, closeness, and betweenness for temporal networks [13]. Takaguchi et al. [14] proposed centrality measures for two-timing nodes at a specific time based on the fastest timing path of the timing nodes (a combination of the node index and time), and found that these two centrality measures are robust to changes in time scale. Ye et al. [15] applied k-shell decomposition based on edges in temporal networks. They proposed a temporal index to represent the importance of nodes in temporal networks, which considers the temporal k-core values of the node itself and its neighbors simultaneously. Qu et al. [16] developed a temporal information-gathering (TIG) process to explore the impact of timing information on the importance of nodes. Ogura et al. [17] proposed the Markov temporal network model, which optimizes the Katz centrality of a given node by continuously adjusting the weights of the edges in the network at the lowest cost. This solves the optimization problem of key node identification. Luo et al. [18] extended centrality based on entropy in static networks to temporal networks from the perspective of information theory. However, these methods [12,14,15,16,17,18] consider only the connection relationships in a time slice.

To completely represent the structural evolution and dynamic process of temporal networks, researchers need to consider the connection relationships between different time slices [10]. Guo et al. [19] proposed the multi-attribute sorting method (TOPSIS) [20]. By calculating the Euclidean distance of different inter-layer coupling indexes to the positive-ideal solution and the negative-ideal solution, TOPSIS ranks the indexes according to the measurement that the results are close to the positive-ideal solution and far away from the negative-ideal solution. Taylor et al. [21] analyzed a multilayer coupling network and established a supra-adjacency matrix (SAM) based on the inter- and intra-layer relationships of a temporal network. Yang et al. [10] believed that the parameter of interlayer relations in SAM ignores discrepancies in the connections of nodes in different layers, so they proposed the similarity-based supra-adjacency (SSAM) method to construct models of temporal networks. Hu et al. [9] constructed an isomorphism rate-based SAM (ISAM) that expresses the coupling relationship between nonadjacent networks. Jiang et al. [22] introduced a time attenuation factor to more accurately describe the coupling strengths of nodes in different time layers to express different inter-layer relationships, and the attenuation-based SAM (ASAM) temporal network model was established.

Although the modeling and results of these aforementioned methods [9,10,21,22] are more realistic and detailed, the representation of intra-layer relationships is relatively simple and does not assess the global importance of nodes. To address these challenges, in this paper, an optimized supra-adjacency matrix (OSAM) temporal network modeling method was established. The OSAM method considers the influence of intra-layer relationships on the importance of nodes, and the weights of edges between adjacent nodes are added to the adjacency matrix of each time layer to express the intra-layer connection relationships. For the coupling relationship between time layers, we propose an improved similarity index based on the real world to express the inter-layer connection relationships of nodes [3]. With the help of the features of directed networks, the OSAM method makes relationship between time layers directional. In addition, a parameter was set, which was calculated by the average of the sum of the eigenvector centrality indices for a node in each layer, and the node importance sorted list was obtained from this index to express the global importance of nodes in temporal networks. To verify the OSAM performance, we applied OSAM to three real networks and compared the experimental results of SAM and SSAM. The simulations on three real networks suggested that OSAM could effectively evaluate the importance of nodes in temporal networks and provides a new idea for modeling temporal networks.

The major contributions are summarized as follows:

1. Considering the influence of intra-layer relationships on the importance of nodes, we defined an adjacency matrix containing node connection strength (represented by the weight of edges) to express the intra-layer relationships.

2. For the construction of supra-adjacency matrix, we combined the directed network’s characteristics with the matrix’s construction, which is reflected in the different inter-layer relationships between adjacent time layers.

3. Using an index (which was calculated by the average of the sum of the eigenvector centrality indices for a node in each layer), we expressed the global importance of nodes in temporal networks.

The rest of this paper is organized as follows: Section 2 provides the description and modeling process of temporal networks. Section 3 presents detailed description of the OSAM temporal network modeling method. Section 4 includes the experimental settings, data description, and analysis of the results. Section 5 presents a summary of this work.

## 2. Preliminaries

Usually, a network can be defined as a tuple G=(V,E), where the nodes form a node set $V=\left\{v_{1},v_{2},\ldots ,v_{n}\right\}$ and the relationships between nodes constitutes an edge set $E=\left\{e_{1},e_{2},\ldots ,e_{H}\right\}$. In temporal networks, if the time length of events between individuals is omitted, and only the initial time at which two individuals interact in a time window is considered, the elements in the edge set E can be described by triples (i, j, t) indicating that node i interacts with node j at time t. We divided the temporal network’s whole period  [t,t+S] into T time windows. The size of each time window is τ=S/T*,* and t equally spaced, nonoverlapping, and continuous time windows { [t, t+τ), [t+τ, t+2τ),…, [t+(T−1)τ, t+S) } can be obtained, so the temporal network was divided into t discrete ordered time-layer networks $G_{1},\;G_{2},\;\ldots ,\;G_{T}$.

The temporal network model based on snapshots can be seen as an extension of the temporal network model based on a static graph, which shows the evolution process of events to a certain extent [23,24]. The multilayer graph temporal network model is able to represent the structural evolution and dynamic process of a temporal network well, and it is generally defined by intra- and inter-layer relationships. An intra-layer relationship corresponds to a snapshot and can be understood as a time-window graph that represents the interaction between nodes. An inter-layer relationship represents a relationship between the corresponding nodes in the adjacent time window and includes only connections between these nodes [11].

Based on the multilayer graph temporal network model, the SAM model established by Taylor is an NT×NT block matrix. The set of ordered time layers of the network was defined as $\;\Gamma =\left\{G_{t}\right\}\;\left(\;t=1,2,\ldots ,T\right)$, where T is the total number of segmented time layers. Then, the SAM model is expressed explicitly as follows:(1)$$A=\left[\begin{array}{cccc} A^{\left(1\right)} & \omega I & 0 & \cdots \\ \omega I & A^{\left(2\right)} & \omega I & \ddots \\ 0 & \omega I & A^{\left(3\right)} & \ddots \\ \vdots & \ddots & \ddots & \ddots \end{array}\right]$$
where $A^{\left(1\right)},\;A^{\left(2\right)},\ldots ,A^{\left(T\right)}$ are the adjacency matrices of the networks in each time slice, they are located on the diagonal of the supra-adjacency matrix and show the intra-layer relationships. $a_{ij}\left(t\right)$ is an element in the adjacency matrix $\;A^{t}$, $a_{ij}\left(t\right)=1$ indicates that node i is connected to node j by an edge in time layer $G_{t}$, and $a_{ij}\left(t\right)=0$ indicates that there is no edge between node i and node j. ωI represents the inter-layer connection relationship; ω is a fixed parameter, where layers become uncoupled when lim→0 and the coupling of layers is strong when lim→∞. I is the N×N identity matrix. The other parts of the supra-adjacency matrix A are represented by 0 because SAM considers only the connection relationships of adjacent time layers.

## 3. Description of the OSAM Temporal Network Modeling Method

### 3.1. Expression of Intra-Layer Relationships

Existing methods [9,10,21,22] have made a series of improvements to inter-layer relationships, but the network in each time slice is only represented by an adjacency matrix; this ignores the impact of intra-layer relationships on the importance of nodes in temporal networks. A node has edges connecting to each neighbor, but the strengths of all edges are not the same; this situation is consistent with the real world, in which a person has different levels of closeness to each friend. To satisfactorily represent intra-layer relationships in each time slice for a temporal network, filling in the weight of each edge between connected nodes at the corresponding position in the adjacency matrix, the weight of an edge is defined as follows:(2)$$w_{ij}=d_{i}+d_{j}$$
where $d_{i}$ and $d_{j}$ indicate the degrees of node i and node j. $w_{ij}\neq 0$ means that node i and node j are connected, and it also expresses the connection strength between the two nodes. The elements in the adjacency matrix are not only 0 or 1 but also the weights of the edges in the slice network. The adjacency matrix after improvement accurately expresses the structure of the network in each time slice and considers the impact of intra-layer relationships on the importance of the nodes in the temporal network.

### 3.2. Expression of Inter-Layer Relationships

To represent inter-layer relationships, most scholars adopt the similarity of nodes between adjacent time snapshots. This paper proposes an improved similarity index for nodes based on ecological competition [3], and the formula is given by Equation (3):(3)$$s_{i}^{\left(t,t+1\right)}=\frac{\sum_{j}a_{ij}\left(t\right)a_{ij}\left(t+1\right)}{\sum_{j}a_{ij}\left(t+1\right)}$$
where $a_{ij}\left(t\right)$ and $a_{ij}\left(t+1\right)$ are elements in the adjacency matrices from the relevant networks $G_{t}$ and $G_{t+1}$ which are adjacent time layer networks. If there is a connection between node i and node j in any time layer network $G_{t}$, $a_{ij}\left(t\right)=1$; otherwise, $a_{ij}\left(t\right)=0$. In fact, Equation (3) shows that there are differences in the similarity between adjacent nodes the similarity between node i and node j is different from that between node j and node i, the node with a larger number of neighbor nodes has a greater influence on the other. The explanation is consistent with intuitive judgment. Taking the ecological competition network as an example, species with similar food sources compete for food, and the competition will be relatively intense. In this process, species with more kinds of food have more choices, so the competition pressure from species with fewer kinds of food is far lower than that from species with more. Therefore, the matrix of adjacent inter-layer connection relationships can be shown as
(4)$$S^{\left(t-1,t\right)}=\left[\begin{array}{cccc} s_{1}^{\left(t-1,t\right)} & 0 & 0 & \cdots \\ 0 & s_{2}^{\left(t-1,t\right)} & 0 & \ddots \\ \vdots & \ddots & \ddots & \ddots \\ 0 & 0 & \ddots & s_{N}^{\left(t-1,t\right)} \end{array}\right]$$

$S^{\left(t-1,t\right)}$ is an N×N diagonal matrix, and $s_{i}^{\left(t-1,t\right)}$ is the improved similarity represented by Equation (3) used to describe the inter-layer similarity of node i; the other parts of matrix $S^{\left(t-1,t\right)}$ are represented by 0.

### 3.3. Modeling the Optimized Supra-Adjacency Matrix

After explaining the intra- and inter-layer relationships, the specific expression of the optimized supra-adjacency matrix is given as follows:(5)$$OSAM=\left[\begin{array}{cccc} A^{\left(1\right)} & S^{\left(1,2\right)} & 0 & \cdots \\ S^{\left(2,1\right)} & A^{\left(2\right)} & S^{\left(2,3\right)} & \ddots \\ 0 & S^{\left(3,2\right)} & A^{\left(3\right)} & \ddots \\ \vdots & \ddots & \ddots & \ddots \end{array}\right]$$
where $A^{\left(1\right)}$, $A^{\left(2\right)}$,..., $A^{\left(T\right)}$ are the adjacency matrices corresponding to T time layers, which include the weights of the edges between each connected node pair used to represent the intra-layer connection relationship. $S^{\left(1,2\right)}$, $S^{\left(2,1\right)}$, $S^{\left(2,3\right)}$, $S^{\left(3,2\right)}$,..., $S^{\left(t-1,t\right)}$ represent the inter-layer relationships between adjacent time layers, and the other parts of the matrices are represented by 0. In particular, this method combines the directed network’s characteristics with the matrix’s construction, which is reflected in the different inter-layer relationships between adjacent time layers. For instance, $S^{\left(1,2\right)}$ is a matrix that indicates the connection relationship between network $G_{1}$ and network $G_{2}$, but $S^{\left(1,2\right)}\neq \;S^{\left(2,1\right)}$, and the difference in the relationship between adjacent layers is more consistent with the real world. The model established by the OSAM method accurately expresses the structure of the temporal network and considers the influence of intra- and inter-layer relationships on the importance of nodes.

### 3.4. Calculation of the Eigenvector Centrality

Eigenvector centrality is an important index for evaluating the importance of network nodes [25], which not only considers the importance of nodes but also comprehensively considers the importance of neighbor nodes. First, we calculated the main eigenvector (the eigenvector corresponding to the maximum eigenvalue) for OSAM $\nu ={\left\{v_{1},v_{2},\ldots ,v_{NT}\right\}}^{T}$. The N(t−1)+ith element of the vector ν indicates the centrality of node i in time layer t, which was recorded as the N×T matrix $W={\left\{\omega_{it}\right\}}_{N\times T}$
(6)$$\omega_{it}=v_{N\left(t-1\right)+i}$$
where $\omega_{it}$ is the element in row i and column t of matrix W and represents the eigenvector centrality of node i in time layer t. This indicator reflects the importance of the nodes in each time slice, but it cannot indicate the global importance of nodes. To address this disadvantage, a parameter was set, which was defined as
(7)$$R_{i}=\frac{\sum_{t=1}^{T}\omega_{it}}{T}$$
where T is the number of time slices, $\omega_{it}$ is the eigenvector centrality of node i in time layer t, and $R_{i}$ is calculated by the average of the sum of the eigenvector centrality indices for the node in each layer to represent the global importance of node i. When the environment around a node change suddenly over time, its importance will also change constantly; $R_{i}$ takes this situation into account and provides a comprehensive evaluation. The list of node global importance is obtained after arranging all $R_{i}$ values from large to small. Taking Figure 1 as an example, the details of the OSAM method are given as follows:

**Step 1**: Calculate the weights of the edges and establish the intra-layer relationship adjacency matrix for the networks in each time slice. Taking network $G_{1}$ as an example, we have
$$\begin{array}{c} w_{13}=d_{1}+d_{3}=3\;\;w_{14}=d_{1}+d_{4}=3\;\\ A^{\left(1\right)}=\left[\begin{array}{cccc} 0 & 0 & 3 & 3\\ 0 & 0 & 0 & 0\\ 3 & 0 & 0 & 0\\ 3 & 0 & 0 & 0 \end{array}\right] \end{array}$$

Similarly, the calculation process of step 1 for $G_{2}$ and $G_{3}$ is as follows:$$\begin{array}{c} w_{13}=d_{1}+d_{3}=3\;\;w_{14}=d_{1}+d_{4}=4\;\;w_{24}=d_{2}+d_{4}=3\;\\ A^{\left(2\right)}=\left[\begin{array}{cccc} 0 & 0 & 3 & 4\\ 0 & 0 & 0 & 3\\ 3 & 0 & 0 & 0\\ 4 & 3 & 0 & 0 \end{array}\right]\\ w_{13}=d_{1}+d_{3}=3\;\;w_{24}=d_{2}+d_{4}=3\;\;w_{34}=d_{3}+d_{4}=4\\ A^{\left(3\right)}=\left[\begin{array}{cccc} 0 & 0 & 3 & 0\\ 0 & 0 & 0 & 3\\ 3 & 0 & 0 & 4\\ 0 & 3 & 4 & 0 \end{array}\right] \end{array}$$

**Step 2**: Calculate the similarity of the nodes in adjacent time slices and establish the inter-layer relationship adjacency matrix for the networks in each time slice. Taking networks $G_{1}$ and $G_{2}$ as examples, we have
$$\begin{array}{c} s_{1}^{\left(1,2\right)}=\frac{2}{2}=1\;s_{2}^{\left(1,2\right)}=\frac{0}{1}=0\;s_{3}^{\left(1,2\right)}=\frac{1}{1}=1\;s_{4}^{\left(1,2\right)}=\frac{1}{2}=0.5\\ S^{\left(1,2\right)}=\left[\begin{array}{cccc} 1 & 0 & 0 & 0\\ 0 & 0 & 0 & 0\\ 0 & 0 & 1 & 0\\ 0 & 0 & 0 & 0.5 \end{array}\right]\\ s_{1}^{\left(2,1\right)}=\frac{2}{2}=1\;s_{2}^{\left(2,1\right)}=0\;s_{3}^{\left(2,1\right)}=\frac{1}{1}=1\;s_{4}^{\left(2,1\right)}=\frac{1}{1}=1\\ S^{\left(2,1\right)}=\left[\begin{array}{cccc} 1 & 0 & 0 & 0\\ 0 & 0 & 0 & 0\\ 0 & 0 & 1 & 0\\ 0 & 0 & 0 & 1 \end{array}\right] \end{array}$$

**Step 3**: The optimized supra-adjacency matrix (OSAM) can be described as
$$\begin{array}{cc} \;\mathrm{OSAM}\; & =\left[\begin{array}{cccccccccccc} 0 & 0 & 3 & 3 & 1 & 0 & 0 & 0 & 0 & 0 & 0 & 0\\ 0 & 0 & 0 & 0 & 0 & 0 & 0 & 0 & 0 & 0 & 0 & 0\\ 3 & 0 & 0 & 0 & 0 & 0 & 1 & 0 & 0 & 0 & 0 & 0\\ 3 & 0 & 0 & 0 & 0 & 0 & 0 & 0.5 & 0 & 0 & 0 & 0\\ 1 & 0 & 0 & 0 & 0 & 0 & 3 & 4 & 1 & 0 & 0 & 0\\ 0 & 0 & 0 & 0 & 0 & 0 & 0 & 3 & 0 & 1 & 0 & 0\\ 0 & 0 & 1 & 0 & 3 & 0 & 0 & 0 & 0 & 0 & 0.5 & 0\\ 0 & 0 & 0 & 1 & 4 & 3 & 0 & 0 & 0 & 0 & 0 & 0.5\\ 0 & 0 & 0 & 0 & 0.5 & 0 & 0 & 0 & 0 & 0 & 3 & 0\\ 0 & 0 & 0 & 0 & 0 & 1 & 0 & 0 & 0 & 0 & 0 & 3\\ 0 & 0 & 0 & 0 & 0 & 0 & 1 & 0 & 3 & 0 & 0 & 4\\ 0 & 0 & 0 & 0 & 0 & 0 & 0 & 0.5 & 0 & 3 & 4 & 0 \end{array}\right] \end{array}$$

**Step 4**: Calculate the eigenvector that corresponds to the largest singular value of the OSAM, determine $R_{i}$, and calculate the sorted list of node importance according to $R_{i}$.

## 4. Experimental Analysis

### 4.1. Data Sets

To verify the effectiveness and accuracy of the OSAM method, three real networks were selected: Enron [26], Emaildept3 [27], and Workspace [28]. The basic statistical features of each network are shown in Table 1.

In Table 1, N is the number of nodes in the network, C is the total number of interactions between nodes, and T represents the number of time layers.

### 4.2. Evaluation Method

In this paper, susceptible-infected (SI) and susceptible-infected-recovered (SIR) spreading models were used to evaluate and compare the results of the OSAM method. NDCG@10 (normalized discounted cumulative gain) was used to evaluate the quality of the sorted list of node importance [29]. We selected SSM and SSAM as the compared methods.

#### 4.2.1. Evaluation Based on the Spreading Model

It is generally believed that the importance of a node is reflected in the ability of messages transmission by a node in the network [30]. Therefore, the information spreading model was used to evaluate the importance of node information transmission in the network. The importance of a node is measured by the total number of network infected nodes in the SI and SIR models. The SI model divides nodes in the network into two states: the susceptible state and the infected state. When a node is susceptible, it can be infected by its neighbor nodes at any time. When a node is infected, it has the ability to infect its neighbor nodes. In the SIR model, the recovered state is added, and an infected node recovers with a certain recovery rate. If the total number of nodes in a network is n, $i_{0}$ nodes in the network are set as the infection sources at the initial time $t_{0}$ and the remaining n−i nodes are uninfected nodes. Then the total number of infected nodes at time i is
(8)$$i\left(t\right)=\frac{n}{\left[1+\left(\frac{n}{i_{0}}-1\right)\exp\left(-p\right)\right]}$$

In the experiment, all nodes were sorted according to the global importance calculated by eigenvector centrality based on OSAM. The nodes with the highest importance in the sorted list were obtained as the infection sources of the spreading model. The numbers of nodes infected at the same time were compared by the SI and SIR models; the SI model was used to evaluate the rationality of OSAM and SIR was used to compare it with other methods.

#### 4.2.2. Evaluation based on Correlation

The normalized discounted cumulative gain (NDCG) is always used to evaluate the quality of the sorted lists. The value range of NDCG is  [0, 1]. The higher the value is, the closer the given sorted list is to the ideal sorted list and the better the method. The NDCG of the top-k nodes in the sorted list was defined as
(9)$$NCDG@k=\frac{DCG@k}{IDCG@k}$$
where IDCG@k is the ideal discounted cumulative gain for the top-k nodes and DCG@k is the discounted cumulative gain, which was defined as
(10)$$DCG@k=\overset{k}{\underset{i=1}{\sum }}\frac{r_{i}}{lb\left(i+1\right)}$$
where $r_{i}$ is the importance score of the *i*-th node in the sorted list.

### 4.3. Experimental Results

#### 4.3.1. Message Transmission Capability of Nodes in a Temporal Network

First, the SI model was used to investigate the message spreading rate of nodes with different rankings in the node importance list obtained by OSAM to illustrate its effectiveness. Specifically, five nodes with the same interval in the node importance sorted list were selected as message sources separately to compare their message spreading rates. If the top nodes had a higher message spreading rate (coverage area under the curve in Figure 2) than the bottom nodes, this means that the nodes with high importance can spread the message to more nodes faster and verifies that the method is effective for the evaluation of node importance.

Figure 2a displays the message spreading rate after selecting the first, 40th, 80th, 120th, and last nodes as the spreading source of the SI model in the Enron network. Due to the long duration of the Enron data set and dense network connections, when a single node was used as the message source, the message could basically cover the whole network (the last node is an isolated node). In addition, the message spreading speed of the first node was significantly faster than those of the lower-ranked nodes. Figure 2b shows the message spreading rate of the first, 25th, 50th, 75th, and last nodes in the Emaildept3 data set. The duration of the Emaildept3 network is relatively short, and messages can quickly cover most networks. In general, the top node has the fastest message spreading rate; the lower the ranking is, the slower the message spreading rate, and the propagation rate of the nodes in the network is generally higher. In Figure 2c, the overall spreading rate of the nodes is not high and the message spreading ability between different nodes is not large, but the nodes with a lower ranking are significantly lower than the nodes that have a higher ranking in terms of message spreading rate and message coverage. The experimental results showed that when the nodes were ranked from high to low according to the node importance obtained by the OSAM method and five nodes were selected as the spreading sources of the SI model, the message spreading rate of the top-ranked nodes was higher than that of the bottom-ranked nodes.

In addition, the SIR model was used to compare the OSAM method with the SAM and SSAM methods to verify the correctness of the model. Specifically, we selected the top 5 nodes in the node importance sorted lists obtained by SAM, SSSM, and OSAM as the important node set S. The top1, top2, top3, top4, top5, and S obtained by the different methods were used as the spreading sources of the SIR model to compare the message coverage of different methods. Under the same number of propagation time steps, the more nodes a message covers, the more accurate the node importance evaluation method. Table 2, Table 3 and Table 4 show the top 1–5 nodes ranked by the SAM, SSAM, and OSAM methods and the important node set S composed of these nodes under the Enron, Emaildept3, and Workspace temporal network data sets.

First, S was taken as the spreading source to investigate the results of message coverage under the three data sets. Figure 3 shows the experimental results. Next, the message coverage of the top 1–5 nodes obtained by the different methods under the SIR model was investigated. Figure 4, Figure 5 and Figure 6 show a comparison of the message coverage of the top 1–5 nodes of the different methods under the SIR model for the Enron, Emaildept3, and Workspace data sets.

Figure 4 presents the comparison results of Top 1–5 under the Enron dataset. Among them, Figure 4a shows that when node 127 was the spreading source of the SIR model, the message coverage was better than that for node 107 and indicates that node 127 is more important than node 107. In the same way, Figure 4c,e shows that node 129 is more important than node 51 and that node 65 is more important than node 92. Obviously, when the important node set S  obtained by the OSAM method was used as the message source, the number of message coverage nodes in the network was higher than with SAM and SSAM.

It can be seen from Table 3 that nodes 49, 60, 25, 23, and 25 were regarded as the top 5 nodes of the Emaildept3 network with the three methods, but the ranking orders were different. As we can see from Figure 5a–e, except for node 49, node 60 performed best, node 25 performed worst, and nodes 35 and 23 performed basically the same. The performance of a single node proves that the ranking order obtained by OSAM is more accurate; Figure 3b also confirms this, as the number of message coverage nodes obtained by OSAM was higher than those of SAM and SSAM.

In the Workspace network, there was little difference in the top 5 nodes calculated by the three methods. Figure 6d,e display that node 38 and node 73 did not perform well, but Figure 6a shows the top node derived from the OSAM method, node 28, performed better. As the most important node, the influence of node 28 was obviously greater. On the whole, Figure 3c shows that the message coverage of the network was slightly higher when the important node set S obtained by OSAM is used as the message source.

In summary, on the three real data sets, the top 1–5 nodes obtained by the OSAM method and the set S of important nodes composed of them had higher message coverages than those of the SAM and SAM methods on the SIR spreading model. This is because this method takes into account the influence of both the intralayer and interlayer relationships on node importance, so it can more accurately identify the important nodes in the temporal network.

#### 4.3.2. Quality of the Node Importance Sorted List

According to SAM, SSAM, and OSAM, a sorted list of node importance was obtained, and the results of the message spreading model were sorted to obtain a standard. The NDCG@10 values of the three methods were calculated for different data sets, and the sorting quality is shown in Table 5.

Table 5 shows that the SAM method had a rough description of the relationship between layers, resulting in the poor quality of the sorted list. Although the SSAM method optimized the relationship between layers and improved the quality of the sorted list to a certain extent, it did not consider the relationships within layers, and the overall performance of NCDG@10 was intermediate. The NDCG@10 of the OSAM method was the highest among the three networks, which indicates that the OSAM method is more accurate for expressing temporal networks. Based on the above analysis, the OSAM method can effectively evaluate the importance of nodes in temporal networks, and the experimental results showed that the OSAM method is superior to other existing methods according to many evaluation methods.

## 5. Conclusions

This paper considered the influence of intra- and inter-layer relationships on nodes’ importance and established an optimized supra-adjacency matrix based on the characteristics of directed networks to model temporal networks precisely. The experimental results showed that compared with SAM and SSAM methods, the optimized supra-adjacency matrix method has good sorted list quality and can accurately identify nodes with strong message propagation capability in the temporal network. When these nodes are used as the source of message propagation, it can improve message coverage and propagation rate. Future studies will explore the length of optimal slicing for the temporal network to improve the identification’s effectiveness.

## Figures and Tables

**Figure 1 entropy-24-01391-f001:**
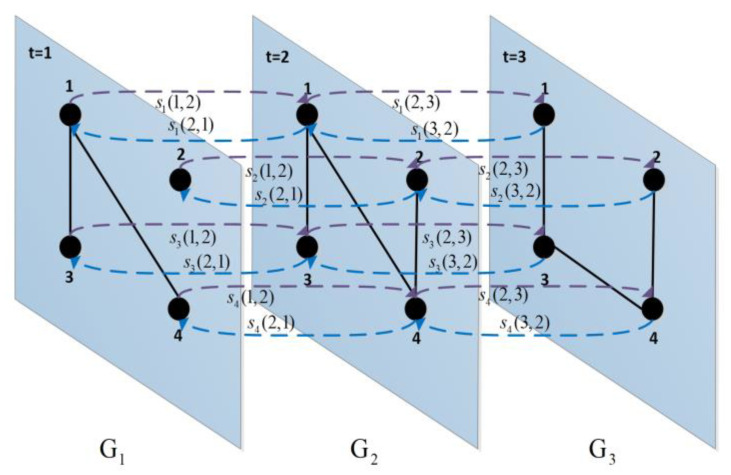
An example of the OSAM model for a temporal network.

**Figure 2 entropy-24-01391-f002:**
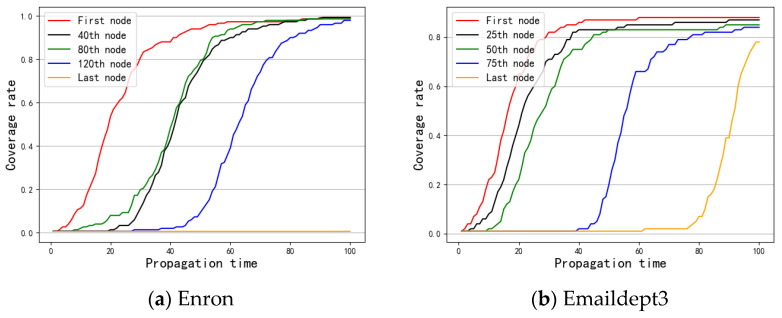
Comparison of network message propagation rate.

**Figure 3 entropy-24-01391-f003:**
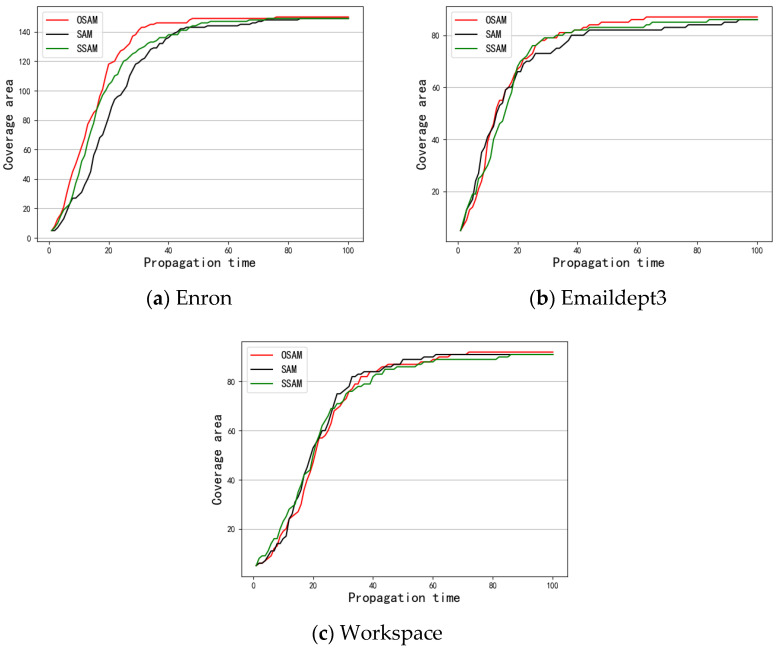
Comparison of network message coverage.

**Figure 4 entropy-24-01391-f004:**
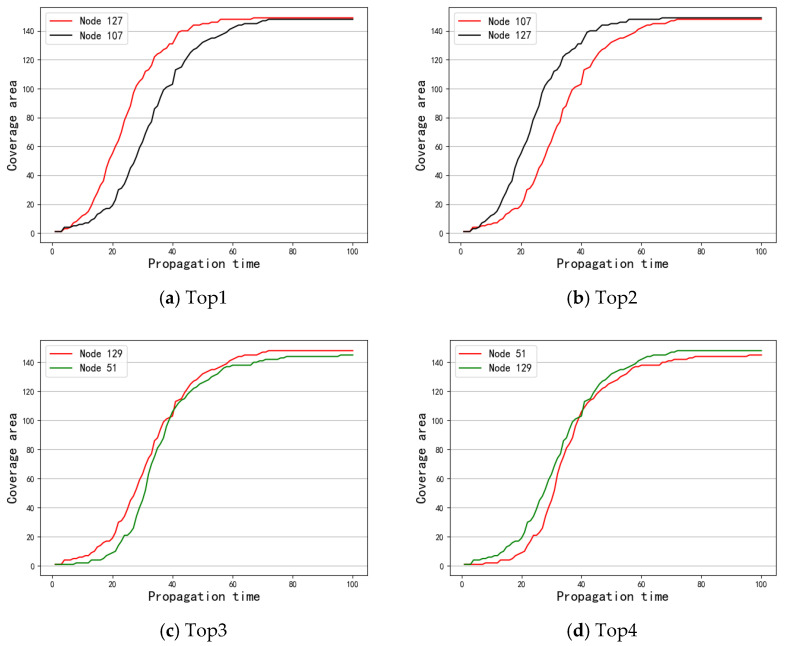
Comparison of different methods top 1–5 under Enron.

**Figure 5 entropy-24-01391-f005:**
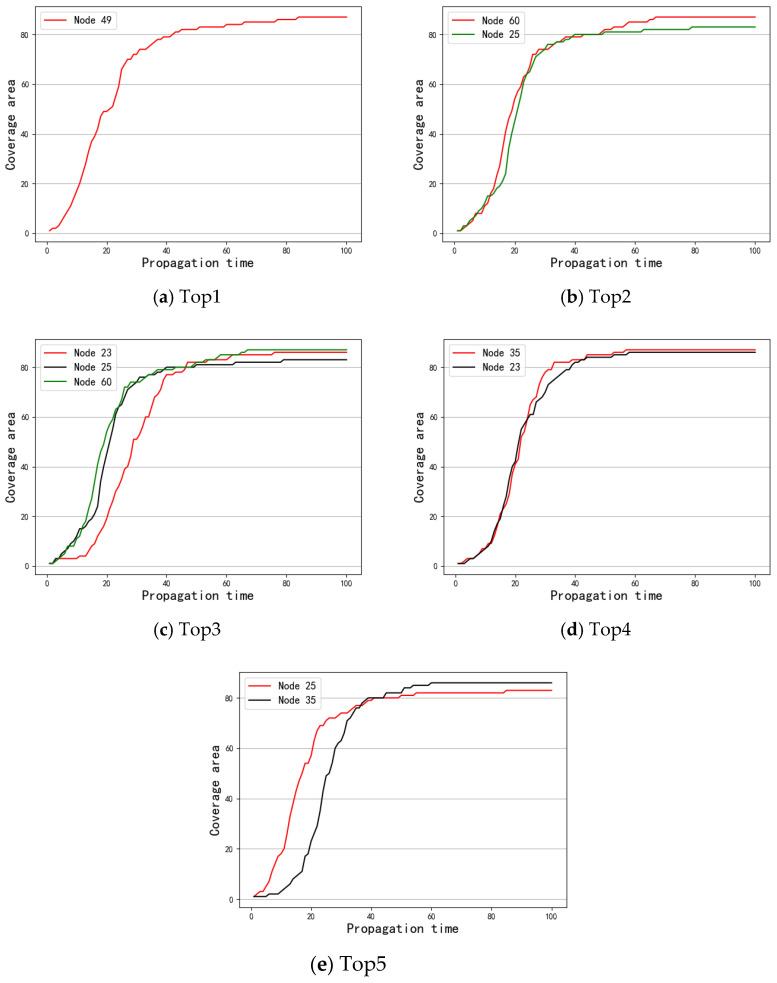
Comparison of different methods top 1–5 under Emaildept3.

**Figure 6 entropy-24-01391-f006:**
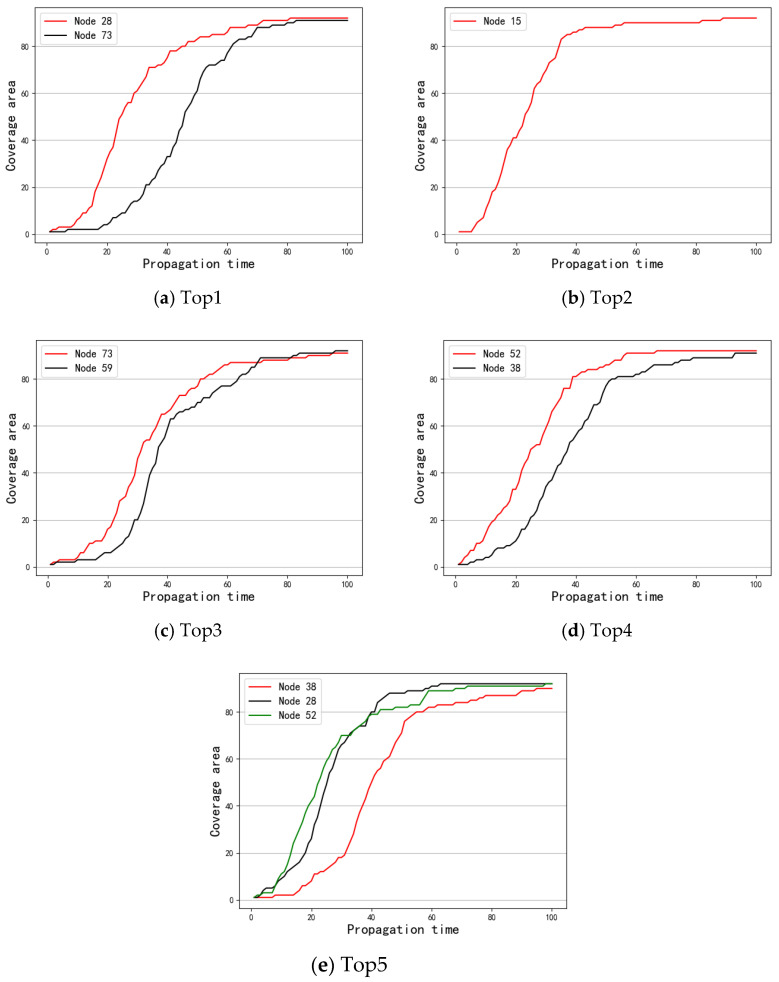
Comparison of different methods top 1–5 under Workspace.

**Table 1 entropy-24-01391-t001:** Basic statistical features of the Enron, Emaildept3, and Workspace networks.

Data Set	N	C	Time Span	T
Enron	151	33,124	1 year	12
Emaildept3	89	12,216	802 days	9
Workspace	92	9287	6/24/–7/3, 2013	10

**Table 2 entropy-24-01391-t002:** Enron node importance rank table.

Method	Top1	Top2	Top3	Top4	Top5	S
SAM	107	127	129	51	92	{107,127,129,51,92}
SSAM	107	127	51	19	91	{107,127,51,19,91}
OSAM	127	107	129	51	65	{127,107,129,51,65}

**Table 3 entropy-24-01391-t003:** Emaildept3 node importance rank table.

Method	Top1	Top2	Top3	Top4	Top5	S
SAM	49	60	25	23	35	{49,60,25,23,35}
SSAM	49	25	60	23	35	{49,25,60,23,35}
OSAM	49	60	23	35	25	{49,60,23,35,25}

**Table 4 entropy-24-01391-t004:** Workspace node importance rank table.

Method	Top1	Top2	Top3	Top4	Top5	S
SAM	73	15	59	38	28	{73,15,59,38,28}
SSAM	73	15	59	38	52	{73,15,59,38,52}
OSAM	28	15	73	52	38	{28,15,73,52,38}

**Table 5 entropy-24-01391-t005:** Sorted list quality.

Enron		Emaildept3		Workspace	
Method	NCDG@10	Method	NCDG@10	Method	NCDG@10
SAM	0.5556	SAM	0.3433	SAM	0.3677
SSAM	0.6074	SSAM	0.4881	SSAM	0.4383
OSAM	**0.6481**	OSAM	**0.6622**	OSAM	**0.5376**

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
