# Peer review of "Node Importance Identification for Temporal Networks Based on Optimized Supra-Adjacency Matrix"

_entropy, 2022, doi:10.3390/e24101391_

Round 1

Reviewer 1 Report

It was a good time for me. There is no suggestion at this moment.

Author Response

We are very grateful to the reviewer for reviewing the paper so carefully. Thank you for your decision and constructive comments on my manuscript. 

Reviewer 2 Report

The paper is well written and presents the method clearly. In addition, the experimental results are interesting.

However, I believe that they should conduct a more in-depth review of methods for determining the importance of nodes in networks. I recommend reading and incorporating the following references:

Ai X, Node Importance Ranking of Complex Networks with Entropy Variation. Entropy 2017, 19, 303.

Dapena A, Iglesia D, Vazquez-Araujo FJ, Castro PM. New Computation of Resolving Connected Dominating Sets in Weighted Networks. Entropy. 2019; 21(12):1174.

On the line 56, it is advisable to indicate the meaning of the acronyms: Technique for Order Preference by Similarity to Ideal Solution (TOPSIS). I also recommend to include a brief description TOPSIS and some references.

The definition of matrix in equation (1) requires to known adjacent of the networks in each time slice. This is a theoretical result that implies knowing "a priori" how the behavior of the network will be. I ask you to indicate in that section or in the simulation section, how to obtain this information.

This issue is also related to the model presented in the equation (8) for infected nodes. It is important to explain whether this is a theoretical or real model.

Reviewer 3 Report

The article proposes a method for identifying important nodes in temporal networks and tests the effectiveness of the method in three real networks. This article is well organized but poorly written. I will suggest major revisions based on the following comments:

1.     The authors should correct the grammar to improve the readability of this article.

2.     The authors should improve the clarity of the figures and use point-line plots instead of line plots to improve the differentiation of different strategies.

3.     The authors should give more description to each figure.

4.     It would be better if the layout of the figures could be readjusted.

5.     Some of the matrix expressions in this article are not standardized, such as Equation (5). The authors should correct the expressions of these matrices.

6.     The author should add more experiments and explain the effectiveness of the OSAM method, especially in the following questions.

1)     The network size is small in the current experiments. What happens if there are more nodes in the network?

2)     Are there synthetic networks to test the effectiveness of the method? In what kinds of networks does the method perform best?

Round 2

Reviewer 3 Report

I do not think the authors have treated my comments carefully.